# The Effect of Fungal Nutraceutical Supplementation on Postoperative Complications, Inflammatory Factors and Fecal Microbiota in Patients Undergoing Colorectal Cancer Surgery with Curative Intent: A Randomized, Placebo-Controlled, Double-Blind Clinical Trial

**DOI:** 10.3390/biomedicines13051185

**Published:** 2025-05-13

**Authors:** Cristina Regueiro, Astrid Irene Diez Martín, Sonia Pérez, Carlos Daviña-Núñez, Sara Zarraquiños, David Remedios, Cristina Alejandra Sánchez Gómez, Sara Alonso Lorenzo, Romina Fernández Poceiro, María Luisa de Castro Parga, Vicent Hernández Ramírez, Arturo Rodríguez-Blanco, Esteban Sinde, Catalina Fernández-de-Ana, Joaquín Cubiella

**Affiliations:** 1Research Group in Gastrointestinal Oncology Disease in Ourense, 32005 Ourense, Spain; astrid.diez@iisgaliciasur.es (A.I.D.M.); joaquin.cubiella.fernandez@sergas.es (J.C.); 2Microbiology and Infectology Research Group, Galicia Sur Health Research Institute (IIS Galicia Sur), 36312 Vigo, Spain; sonia.maria.perez.castro@sergas.es (S.P.);; 3Microbiology Department, Complexo Hospitalario Universitario de Vigo (CHUVI), SERGAS, 36312 Vigo, Spain; 4Faculty of Biology, Universidade de Vigo, 36312 Vigo, Spain; 5Department of Gastroenterology, Hospital Universitario de Ourense, Centro de Investigación Biomédica en Red de Enfermedades Hepáticas y Digestivas (CIBEREHD), 32005 Ourense, Spain; sara.zarraquinos.martinez@sergas.es (S.Z.); david.rafael.remedios.espino@sergas.es (D.R.); cristina.alejandra.sanchez.gomez@sergas.es (C.A.S.G.); 6Department of Gastroenterology, Complexo Hospitalario Universitario de Vigo, 36312 Vigo, Spain; sara.alonso.lorenzo@sergas.es (S.A.L.); maria.luisa.de.castro.parga@sergas.es (M.L.d.C.P.); vicent.hernandez.ramirez@sergas.es (V.H.R.); 7Research Group in Gastroenterology, Galicia Sur Health Research Institute, 36312 Vigo, Spain; romina.fernandez@iisgaliciasur.es; 8Hifas da Terra S.L., 36154 Pontevedra, Spain; arturo.rodriguez@hifasdaterra.com (A.R.-B.); esteban.sinde@hifasdaterra.com (E.S.); c.fdap@hifasdaterra.com (C.F.-d.-A.)

**Keywords:** colorectal cancer, fungal nutraceutical supplementation, postoperative complications, gut microbiota

## Abstract

**Background/Objectives:** The combination of different fungal extracts could be beneficial to cancer patients due to their role in gut microbiota modulation and anti-inflammatory activity. The study aimed to evaluate whether fungal extract supplementation reduces postsurgical complications in patients with colorectal cancer undergoing curative surgery. **Methods:** Patients were randomized to receive the nutraceutical Micodigest 2.0 or a placebo until surgery. Surgical complications were evaluated using the Clavien-Dindo classification. We also assessed the effect of the nutraceutical on gut microbiota composition, inflammatory response, nutritional status, and quality of life. A subanalysis based on surgery type (robotic vs. non-robotic) was performed. **Results:** We included 46 patients who met the inclusion criteria, with 27 randomized to the intervention group and 19 to the placebo group, receiving treatment for three (2–4) weeks. Non-robotic surgery was performed in 35 (76.1%) patients. We found non-significant differences in postoperative complications (Micodigest 2.0: 25.9%, placebo: 26.3%; *p* = 0.9). In non-robotic surgery, we identified a non-significant reduction in postoperative complications (Micodigest 2.0: 25.0%, placebo: 36.4%; *p* = 0.7), as well as a significant increase in lymphocyte levels and a reduction in the neutrophil-to-lymphocyte ratio (*p* = 0.02). Micodigest 2.0 supplementation was also associated with significant changes in gut microbiota composition, as indicated by a decreased relative abundance of the phyla Firmicutes (*p* = 0.004) and Actinobacteria (*p* = 0.04). **Conclusions:** Micodigest 2.0 supplementation was associated with non-significant reductions in postoperative complications and significant modifications in gut microbiota composition. **Limitations:** The trial did not reach the calculated sample size.

## 1. Introduction

Colorectal cancer (CRC) is one of the most prevalent forms of cancer in developed countries. Epidemiological data indicate that CRC is the third most common cancer in terms of incidence and the second leading cause of cancer-related mortality [1]. The majority of CRC cases are suitable for surgical resection with curative intent, with success rates ranging from 67% to 92%, depending on the tumor stage [2]. Despite significant advancements in surgical techniques, colorectal surgery remains associated with complications that can be potentially life-threatening [3]. Postoperative complications occur in up to one-third of patients undergoing colorectal procedures, with 15% experiencing these complications within the first month after discharge and an additional 25% within the first year following surgery [3,4]. Major complications include wound infections, anastomotic leakage, ileus, and bleeding [5].

The primary causes of postoperative complications are immune dysfunction and immunosuppression triggered by surgical stress [6]. Additionally, postoperative complications are closely linked to gut microbiota dysbiosis [7]. Mechanical bowel preparation and antibiotic prophylaxis, commonly used in colorectal surgery, significantly impact the diversity and composition of the gut microbiota. Mechanical preparation reduces non-pathogenic bacteria such as Bifidobacteria and Lactobacilli while increasing pathogenic species such as *Escherichia coli* and *Staphylococcus* spp. [8]. Similarly, antibiotic prophylaxis and surgical stress alter microbiota diversity and relative abundance [8,9].

In this context, probiotics and synbiotics have shown potential in reducing surgical complications due to their ability to regulate immune responses and maintain gut microbial balance [10,11]. Preoperative administration of immune-modulating nutrition, including prebiotics and synbiotics, has been shown to reduce the risk of postoperative infections and shorten hospital stays, without significantly affecting other surgical outcomes or mortality [12,13]. A recent meta-analysis of 34 randomized clinical trials confirmed that probiotics and synbiotics reduce infectious complications during hospitalization, although they have no significant effect on mortality or non-infectious complications [14].

Fungal polysaccharides, particularly β-glucans, have gained attention for their role in modulating gut microbiota and immune function. β-glucans can reduce pathogenic bacteria and promote the growth of beneficial microorganisms. Extracts from certain basidiomycetes, such as *Ganoderma lucidum* [15], *Pleurotus eryngii* [16], and *Hericium erinaceus* [17], have demonstrated prebiotic activity in animal models. Moreover, *Pleurotus ostreatus*, *P. eryngii*, and *H. erinaceus* have shown bifidogenic, lactogenic, and short-chain fatty acid production effects in fecal inocula from elderly donors [18].

Beyond their prebiotic effects, fungal polysaccharides exhibit anti-inflammatory properties. Polysaccharides extracted from *Ganoderma lucidum* and *Lentinula edodes* demonstrate immunomodulatory effects in colitis models, promoting nitric oxide, tumor necrosis factor-alpha (TNF-α), and interleukin-6 (IL-6) production [19]. Additionally, basidiomycete extracts improve immune function in inflammatory bowel disease, while *G. lucidum* reduces pro-inflammatory cytokines in CRC patients [20,21]. Combining fungal extracts may optimize immune modulation, enhancing intracellular signaling and immune responses [22,23].

Micodigest 2.0 is a nutraceutical developed by Hifas da Terra in 2016, containing nine fungal extracts with a favorable safety profile: *Ganoderma lucidum* (Ganozumib^®^), *Agaricus blazei*, *Grifola frondosa*, *Hericium erinaceus* (Herizumib^®^), *Lentinula edodes* (Lentizumib^®^), *Cordyceps sinensis*, *Inonotus obliquus*, *Pleurotus ostreatus*, and *Polyporus umbellatus*. Given the potential benefits of fungal polysaccharides, we hypothesized that Micodigest 2.0 could reduce postoperative complications in CRC patients undergoing surgery with curative intent.

## 2. Materials and Methods

### 2.1. Study Design and Patient Enrollment

A randomized, double-blind, placebo-controlled, prospective study was conducted (ClinicalTrials.gov Identifier: NCT04821258) [24]. Patients were recruited from the Clinical University Hospital of Ourense and the Clinical University Hospital of Vigo. CRC patients who were candidates for surgical resection with curative intent and met the inclusion and exclusion criteria were enrolled. The inclusion criteria were: (1) American Society of Anesthesiologists (ASA) physical status classification < 3, (2) age between 18 and 85 years, (3) Eastern Cooperative Oncology Group (ECOG) performance status 0–2, (4) preserved cognitive function, and (5) written informed consent after reviewing the study information sheet. Exclusion criteria included: (1) receipt of neoadjuvant therapy, (2) concomitant malignancy, (3) allergy to the nutraceutical or presence of malabsorption syndrome, (4) mental health disorders, (5) active infection or antibiotic therapy within the past month, and (6) prior colorectal surgery.

Eligible patients were randomized to receive either the nutraceutical or placebo prior to hospital admission. Participants were followed for 4–6 weeks until surgery. The study was conducted in accordance with the Standard Protocol Items: Recommendations for Interventional Trials (SPIRIT) guidelines [25] and was approved by the Clinical Research Ethics Committee of Galicia (2021/036). All participants provided written informed consent, and data privacy was strictly maintained. The final manuscript adheres to the Consolidated Standards of Reporting Trials (CONSORT) guidelines [26].

### 2.2. Nutraceutical and Placebo Treatment

The nutraceutical, Micodigest 2.0, was provided by Hifas da Terra in the form of 30 capsules and 300 mL of syrup. The syrup contained nine organic mushroom extracts from *Ganoderma lucidum* (Ganozumib^®^), *Agaricus blazei*, *Grifola frondosa*, *Hericium erinaceus* (Herizumib^®^), *Lentinula edodes* (Lentizumib^®^), *Cordyceps sinensis*, *Inonotus obliquus*, *Pleurotus ostreatus*, and *Polyporus umbellatus*, along with *Myrciaria dubia*, purified water, raw agave, and natural flavoring. The capsules contained a plant-based shell, *Lactobacillus brevis*, *Lactobacillus plantarum*, magnesium stearate, silicon dioxide, and an extract from *G. lucidum*. Micodigest 2.0 has been approved as a dietary supplement since 2016 and has not been associated with any adverse effects (AEs). The placebo, which contained no fungal extracts, was matched in taste and texture to the nutraceutical and was manufactured by the same company. Both the nutraceutical and placebo were stored at room temperature, protected from light.

### 2.3. Intervention Monitoring

Eligible patients were randomly assigned to either the nutraceutical or placebo group. Patients in the nutraceutical group were instructed to take the supplement orally according to the following schedule: 10 mL of syrup and one capsule per day for the first seven days, followed by an increase to 20 mL of syrup and two capsules per day. The placebo group followed the same regimen. The intervention lasted approximately 4–6 weeks, concluding on the date of surgery. Fecal and blood samples were collected at baseline (before the intervention) and after the intervention period. Blood samples were analyzed for routine hematological parameters, biochemical markers, and inflammatory biomarkers. Fecal samples were stored at −20 °C until microbiome analysis was conducted. Nutritional status and quality of life were assessed before and after the intervention using the Patient-Generated Subjective Global Assessment (PG-SGA) [27] and the SF-36 questionnaire [28], respectively. Adherence to the treatment protocol and the occurrence of AEs were monitored through weekly telephone follow-ups. AEs and their severity were documented according to the Common Terminology Criteria for Adverse Events (CTCAE), version 5.0 [29]. AEs were considered serious if they were classified as ≥III. The treatment schedule and sample collection timeline are illustrated in Figure 1.

### 2.4. Assessment of Postoperative Complications

Postoperative complications were evaluated using the Clavien-Dindo classification (CDC) [30]. This system categorizes complications into five grades based on severity: Grade I includes any deviation from the normal postoperative course that does not require pharmacological treatment or surgical, endoscopic, or radiological intervention. Permitted treatments include antiemetics, antipyretics, analgesics, diuretics, electrolytes, and physiotherapy. Grade II complications require pharmacological treatment beyond that allowed for Grade I, including blood transfusions and total parenteral nutrition. Grade III complications necessitate surgical, endoscopic, or radiological intervention (Grade IIIa: without general anesthesia; Grade IIIb: requiring general anesthesia). Grade IV complications are life-threatening and require intensive or intermediate care (Grade IVa: single-organ dysfunction, including dialysis; Grade IVb: multi-organ dysfunction). Grade V represents death following surgery.

### 2.5. Microbiota Characterization

Fecal samples were collected in sterile, dry containers and stored by patients at 4 °C until transported to the laboratory, where they were frozen at −80 °C within 48 h of collection. DNA was extracted using the QIAsymphony PowerFecal Pro DNA Kit (QIAGEN, Hilden, Germany). Gut microbiota characterization was performed using Illumina DNA Prep (Illumina Inc., San Diego, CA, USA), followed by shallow shotgun sequencing [31] on an Illumina NextSeq sequencer (P2 cartridge, 2 × 150 bp). Sequencing data were processed using the SqueezeMeta pipeline and the SQMtools R package in R Studio (version 4.4.2) [32]. Species with low prevalence (<20%) and low abundance (<100 reads) were filtered out. Alpha diversity indices (Shannon, Simpson, Observed, and Chao1) and beta diversity measures (Bray-Curtis and Jaccard distances) were calculated using the MicrobiomeStat R package. A paired analysis was conducted to compare pre- and post-intervention data, and an independent analysis was performed based on the presence or absence of surgical complications. Alpha diversity was analyzed using generalized linear models. Principal Coordinates Analysis (PCoA) and PERMANOVA were applied for beta diversity visualization and statistical testing. Feature-level analysis was conducted using linear discriminant analysis [33] with the MicrobiomeStat package at various taxonomic levels. Statistical significance was defined as an unadjusted *p*-value < 0.05. Correlations between microbial taxa and clinical blood parameters were assessed using Pearson’s correlation test.

### 2.6. Sample Size Calculation

The sample size was calculated based on an expected complication rate of 40% in the control group, with a hypothesized 50% reduction in the intervention group considered clinically relevant [3]. Assuming a β error of 0.20 (80% power) and an α error of 0.05, a total of 64 patients per group was required. To account for a 10% dropout rate, the final sample size was set at 144 patients (72 per group). The sample size calculation was performed using Ene 3.0 statistical software (GlaxoSmithKline SA, London, UK).

### 2.7. Statistical Analysis

Continuous variables were presented as medians with interquartile ranges (IQR) or means with standard deviations (SD), depending on data distribution. Categorical variables were expressed as frequencies and percentages. Categorical variables were compared using the chi-square test or Fisher’s exact test, as appropriate.

Continuous variables were analyzed using *t*-tests for normally distributed data and Mann-Whitney U tests for non-normally distributed data. A *p*-value < 0.05 was considered statistically significant. No correction for multiple comparisons was applied. So, results should be interpreted as exploratory.

Specific Analyses:Postoperative Complications: The complication rate was compared between the two groups, stratified by severity and type of complication, using chi-square tests. A subgroup analysis was performed based on the type of surgery (robotic vs. non-robotic).Adverse Events (AEs): AEs were documented and compared between groups using chi-square tests.Inflammatory Markers: Differences between groups were analyzed using *t*-tests (for normally distributed variables) or Wilcoxon tests (for non-normally distributed variables).Dietary Patterns and Quality of Life: Changes in qualitative variables were assessed using McNemar’s test, while quantitative data were analyzed with paired *t*-tests.

## 3. Results

### 3.1. Patient’s Baseline Characteristics

Between June 2021 and October 2023, 150 patients who met the inclusion criteria were invited to participate in the study. Of these, 69 initially expressed interest, and 46 patients ultimately consented to participate. The median age of the included patients was 67.0 years (IQR: 62.3–72.8), with women representing 50.0% of the cohort. Recruitment was suspended due to a low inclusion rate. Following randomization, 27 patients were allocated to the nutraceutical group and 19 to the placebo group (Figure 2). The baseline characteristics of both groups are presented in Table 1. There were no significant differences in gender, age, or other demographic variables between the groups. Regarding CRC diagnosis, the majority of patients were diagnosed following a positive fecal immunochemical test (50.0%), with no significant differences observed between groups. Similarly, tumor location did not differ significantly between groups, with most tumors located in the distal colon (54.3%). There were no significant differences in body mass index (BMI) or other anthropometric measures. Routine blood parameters, including inflammatory cytokines, were also comparable between the two groups. These findings indicate that the nutraceutical and placebo groups were homogeneous at baseline.

### 3.2. Association of Nutraceutical Supplementation with Postoperative Complications

As shown in Table 2, patients were monitored for a median of three weeks (IQR: 2–4) prior to surgery. The follow-up duration was consistent between the nutraceutical and placebo groups (*p* = 0.9), with no significant differences observed. During this period, no infections or antibiotic administrations were reported. AEs were reported in 15 patients (32.6%) during follow-up, with 7.1% classified as serious (≥III). The incidence of AEs did not differ significantly between the nutraceutical and placebo groups (25.9% vs. 42.1%, *p* = 0.2).

The most common surgical procedures were right hemicolectomy (30.4%) and sigmoidectomy (30.4%). Other procedures included left hemicolectomy (4.3%), low anterior resection (10.9%), rectosigmoid resection (8.7%), segmental colectomy (6.5%), and other types of resections (8.7%). Most surgeries were non-robotic (76.1%) (Table 2, Appendix A). In terms of surgical approach, 29 patients underwent laparoscopic surgery (82.9%), four required conversion to open surgery (11.4%), and two underwent primary open surgery (5.7%). Additionally, 11 patients (23.9%) underwent robotic-assisted surgery.

Postoperative complications occurred in 26.1% of patients (Figure 3A). The majority were minor (84.4%), classified as Grade I (50%) or Grade II (50%), according to the Clavien-Dindo (CD) classification. The remaining complications were Grade III, indicating greater severity. We found non-significant differences in postoperative complications (Micodigest 2.0: 25.9%, placebo: 26.3%; *p* = 0.9). In the placebo group, most complications were Grade I or II, with one patient experiencing a Grade III complication (5.2%). In the nutraceutical group, only one patient had a Grade III complication (3.7%), while the rest were classified as Grade I or II (Figure 3A). In the subgroup analysis based on surgical technique, no significant associations were found, both in the robotic surgery subgroup (33.4% vs. 12.5%, *p* = 0.4; Figure 3B) and in patients who underwent non-robotic surgery, (25.0% vs. 36.4%, *p* = 0.7; Figure 3B).

### 3.3. Association of Nutraceutical Supplementation with Inflammatory Biomarkers, Nutritional Status, and Quality of Life

At the end of the follow-up period, there were no statistically significant differences in BMI, fat mass, or muscle mass between the placebo and nutraceutical groups (Table 3). The comparable nutritional status of both groups was also reflected in the PG-SGA survey results, with the majority of patients classified as well-nourished. Similarly, quality of life, as assessed by the SF-36 questionnaire, showed no significant differences between the placebo group and the nutraceutical-treated group. No notable differences were observed in any of the SF-36 domains, including physical functioning, role limitations due to physical health, bodily pain, general health, vitality, social functioning, role limitations due to emotional problems, and mental health (Table 2). Routine blood parameters were also comparable between the two groups at the end of the follow-up period (Table 2). Inflammatory markers, including IL-6, IL-10, and TNF-α, showed no significant differences between the placebo and nutraceutical groups (Table 3). In the subgroup of patients undergoing non-robotic surgery, nutritional status and quality of life outcomes were similar between those treated with the placebo and those treated with the nutraceutical (Appendix A). However, a statistically significant difference was observed in laboratory measurements: lymphocyte levels were significantly higher in the nutraceutical group compared to the placebo group [1735 (1213–2035)/µL vs. 1170 (953–1558)/µL; *p* = 0.02, Appendix A]. Additionally, the neutrophil-to-lymphocyte ratio was significantly lower in the nutraceutical group [2.02 (1.26–2.62) vs. 2.98 (2.34–4.70); *p* = 0.02, Appendix A].

### 3.4. Effect of Nutraceutical Supplementation on Fecal Microbial Diversity

Fecal samples were collected at baseline and at the end of the intervention (Figure 1). A total of 45 patients provided a baseline sample—19 from the placebo group and 26 from the nutraceutical group. Among these patients, 36 samples provided a second sample at the end of the treatment, including 13 from the placebo group and 23 from the nutraceutical group. Therefore, microbial diversity analysis was performed for 36 of the total 47 recruited patients. There were no significant differences in alpha diversity within either the placebo or nutraceutical groups when comparing baseline and post-treatment samples (Appendix A). Additionally, no significant differences in alpha diversity were observed between the placebo and nutraceutical groups at either time point (Appendix A, *p* = 0.6). However, beta diversity analysis based on the Jaccard distance revealed a statistically significant difference between the placebo and nutraceutical groups, although substantial overlap was observed between the samples from both groups (Appendix A, *p* = 0.006).

At the phylum level, Actinobacteria, Bacteroidetes, Firmicutes, and Proteobacteria accounted for over 80% of the total bacterial composition in both groups (Appendix A). The most significant differences between the placebo and nutraceutical groups were observed in the Firmicutes (*p* = 0.004, Figure 4A) and Actinobacteria (*p* = 0.04, Appendix A) phyla. At the family level, the abundance of Muribaculaceae (*p* = 0.01), Enterobacteriaceae (*p* = 0.01), Eggerthellaceae (*p* = 0.02), and Eubacteriales Family XIII Incertae Sedis (*p* = 0.03) significantly increased in the Micodigest 2.0 group compared to the placebo group (Figure 4B), while the abundance of Desulfovibrionaceae significantly decreased (*p* = 0.02, Figure 4B). At the genus level, *Parasutterella* sp. (*p* = 0.02) and *Mediterraneibacter* sp. (*p* = 0.03) showed a significant decrease in the Micodigest 2.0 group compared to the placebo group (Figure 4C). In the subgroup of patients who underwent non-robotic surgery, similar outcomes were observed (Appendix A). A total of 26 samples were collected from this subgroup—6 from the placebo group and 20 from the nutraceutical group. Patients supplemented with Micodigest 2.0 exhibited a decrease in the Firmicutes (*p* = 0.02), Actinobacteria (*p* = 0.03), and Bacteroidetes (*p* = 0.02) phyla compared to those in the placebo group (Appendix A). At the family level, patients in the placebo group showed an increased abundance of Muribaculaceae (*p* = 0.01), while patients in the nutraceutical group exhibited a significant increase in Desulfovibrionaceae abundance (*p* = 0.04) (Appendix A). Finally, at the genus level, only *Sanguibacteroides* sp. showed a significant increase in abundance in the placebo group (*p* = 0.003) (Appendix A).

## 4. Discussion

The results of this study indicate that the nutraceutical Micodigest 2.0 was associated with a non-significant reduction in postoperative complications in patients with colorectal cancer (CRC) undergoing surgery with curative intent. Nevertheless, further studies with larger sample sizes are required to more definitively determine the role of this nutraceutical in reducing postoperative complications associated with CRC surgery. Overall, Micodigest 2.0 had no significant impact on the nutritional status, quality of life, or inflammatory markers of CRC patients undergoing curative treatment. However, in the majority of recruited patients (76.1%) who underwent non-robotic surgery, a statistically significant increase in lymphocyte levels and a statistically significant reduction in the neutrophil-to-lymphocyte ratio were observed in the nutraceutical group compared to the placebo group. With regard to gut microbiota, the supplementation with the nutraceutical was associated with statistically significant modifications in microbial diversity, as well as a reduction in the relative abundance of the phyla Firmicutes and Actinobacteria compared to the placebo group. Further studies with larger cohorts are required to confirm these findings and to gain a deeper understanding of the relationship between this nutraceutical, gut microbiota regulation, and other clinical outcomes in CRC patients undergoing curative resection.

Recent studies have highlighted the potential of fungi as beneficial adjuncts in the treatment of various cancers. Medicinal mushrooms have demonstrated the potential to prevent lymph node metastasis, prolong overall survival, reduce chemotherapy-induced side effects (e.g., diarrhea, vomiting), modulate the immune system, and help maintain immune function in certain types of cancer [34,35]. For instance, Lentinula edodes, present in the nutraceutical, has shown anti-cancer properties in CRC patients [34]. Moreover, Macharia et al. demonstrated that mushrooms may have a substantial positive impact on the treatment of CRC patients [36]. A previous meta-analysis of studies in cancer patients supplemented with the medicinal mushroom *Ganoderma lucidum*, present in the nutraceutical, found that patients receiving *G. lucidum* alongside chemotherapy or radiotherapy were more likely to respond positively compared to those receiving chemotherapy or radiotherapy alone. Furthermore, immune function indicators suggested that *G. lucidum* increased the percentages of CD3, CD4, and CD8 T lymphocytes [37]. Similar trends indicating immune modulation by *G. lucidum* extract were observed specifically in advanced CRC, where increased levels of CD3, CD4, CD8, and CD56 lymphocytes were reported [20]. Consistent with these previous findings for Ganoderma lucidum, Micodigest 2.0, a combination of mushroom supplements including Ganoderma lucidum extract Ganozumib, showed a statistically significant increase in lymphocyte levels compared to placebo in the subgroup of patients undergoing non-robotic surgery in our study, representing 76.1% of the total.

In addition, the nutraceutical showed a statistically significant decrease in the neutrophil-to-lymphocyte ratio compared to placebo in the subgroup of patients undergoing non-robotic surgery in our study. The reduction of the neutrophil-to-lymphocyte ratio plays a critical role in modulating the inflammatory response, reflecting a shift from a pro-inflammatory state to a more regulated immune environment. An elevated neutrophil-to-lymphocyte ratio has been associated with poor outcomes in various inflammatory and cardiovascular conditions, indicating heightened neutrophil-driven inflammation coupled with relative lymphopenia, a marker of immune suppression [38,39]. Conversely, a decreased neutrophil-to-lymphocyte ratio suggests a controlled inflammatory response, promoting tissue repair and reducing the risk of chronic inflammation-related damage. The prognostic value of this biomarker has been demonstrated in conditions such as sepsis, cancer, and COVID-19, underscoring its utility in clinical decision-making [40,41].

In a recently published systematic review, elevated preoperative neutrophil-to-lymphocyte ratio values were significantly correlated with severe complications in 13 out of 19 studies (68.4%), with a cutoff value varying between 2.3 and 4. Moreover, in this systematic review, several studies found that preoperative neutrophil-to-lymphocyte ratio was an independent risk factor for various adverse outcomes when multivariate analysis was performed. In the long term, higher neutrophil-to-lymphocyte ratio values were associated with lower overall survival and disease-free survival values [42]. In our study, we identified a non-significant reduction in the ratio across the entire patient cohort, which reached statistical significance in patients who underwent non-robotic surgery, and thus experienced greater surgical trauma. Our results are consistent with previously published data. However, we were unable to confirm an association with clinically relevant outcomes; therefore, further research is needed to determine whether this ratio can serve as a predictor of postoperative evolution following colorectal surgery.

To the best of our knowledge, this is the first study to evaluate the role of fungi in reducing complications associated with colorectal surgery. To date, only studies assessing the effectiveness of probiotics in reducing postoperative complications in CRC patients have been conducted [43]. Our research did not identify a significant benefit of mushroom-based supplementation in reducing postoperative complications associated with CRC surgery. However, positive trends were identified in patients undergoing non-robotic surgery. Given the growing body of evidence supporting their therapeutic effects, further research is warranted to elucidate the potential benefits of mushroom-based nutraceuticals in this context.

Postoperative complications following colorectal surgery have been linked to alterations in gut microbiota [7]. Kong et al. evaluated fecal samples from 43 CRC patients collected before and after surgery, finding significant shifts in fecal microbiota composition postoperatively [44]. Our results indicated a reduction in the relative abundance of Actinobacteria and Firmicutes in patients treated with Micodigest 2.0. The nutraceutical-treated group also demonstrated reduced growth of pathogenic bacteria compared to the placebo group. These findings support the hypothesis that specific fungal extracts can play a pivotal role in regulating gut microbiota, potentially inhibiting the growth of pathogenic organisms. Moreover, these findings are consistent with previous studies on probiotics. A decrease in Firmicutes was observed in CRC patients treated with probiotics prior to surgery [45]; similarly, Zheng et al. reported a reduction in Firmicutes and Actinobacteria in patients treated with probiotics following partial gastrectomy [46]. The information available on the impact of these modifications on the risk of postoperative complications is scarce. Nevertheless, the relative abundance of the phyla may affect the prognosis of CRC patients. In this sense, patients with a high abundance of Proteobacteria, Firmicutes, or Bacteroidetes have significantly improved survival. In contrast, Actinobacteria do not affect survival [47].

The findings in the subgroup of patients undergoing non-robotic surgery warrant further investigation in larger studies to confirm these results. The use of robotic surgery has increased in recent years and has been shown to be both safe and effective for CRC patients undergoing curative procedures [48]. Robotic surgery is associated with a reduction in surgical complications [49], while open surgery is linked to a higher rate of postoperative complications [50]. In our study, only two patients underwent open surgery, and neither experienced complications. Additional studies comparing the outcomes of different surgical approaches are needed to determine whether fungal-based treatments offer specific benefits in certain surgical contexts.

Unfortunately, our study had notable limitations. Despite an inclusion period of almost three years, we did not reach the required sample size, achieving less than 50% of the estimated target. Additionally, we were unable to recruit an equal number of patients in the placebo and nutraceutical groups, which may introduce bias. Moreover, the incidence of postoperative complications was lower than anticipated (26.1% vs. the expected 40%). This lower event rate may explain the absence of significant differences, with relevant findings limited to the subgroup undergoing non-robotic surgery. Nevertheless, this is the first study to assess the use of a mushroom-based nutraceutical in reducing postoperative complications in CRC patients undergoing surgery with curative intent. Importantly, it is a randomized controlled trial, which remains the gold standard for evaluating the efficacy of new therapeutic interventions. The study also included data collected at both baseline and the end of the intervention, allowing us to confirm the absence of pre-existing differences between groups prior to randomization.

Finally, our results indicate that treatment with Micodigest 2.0 did not increase the incidence of adverse events during follow-up. Given the favorable safety profile of the treatment and the potential therapeutic benefits of mushroom-based supplements, further research is needed to confirm their health benefits and potential role in clinical practice.

## 5. Conclusions

In conclusion, the nutraceutical was associated with a non-significant reduction in postoperative complications in patients with colorectal cancer undergoing curative surgery. Regarding the gut microbiota, Micodigest 2.0 supplementation led to statistically significant changes in the composition of the bacterial community at the phylum, family, and genus levels. The treatment also appeared to have a beneficial effect by reducing the abundance of pathogenic bacteria in the gut of CRC patients undergoing colorectal surgery. No significant associations were found between the fungal-based treatment and nutritional status, quality of life, or inflammatory parameters. However, evidence of immune modulation was observed, with a significant increase in lymphocyte counts and a significant decrease in neutrophil-to-lymphocyte ratios in a subgroup of 76.1% of patients undergoing non-robotic surgery. The findings of this study are limited by the small sample size, and further research with larger cohorts is required to confirm the potential health benefits of this treatment. The modulation of the gut microbiota by fungal polysaccharides, as observed in this and previous studies, represents a promising avenue for future research and potential clinical applications.

## Figures and Tables

**Figure 1 biomedicines-13-01185-f001:**
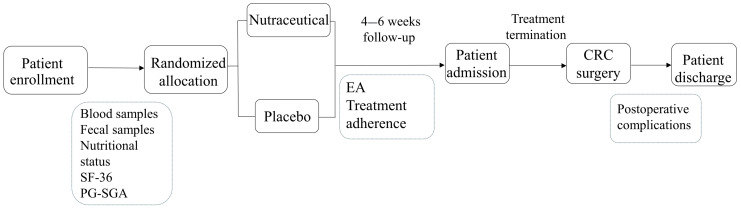
Flow diagram showing the schedule of the study.

**Figure 2 biomedicines-13-01185-f002:**
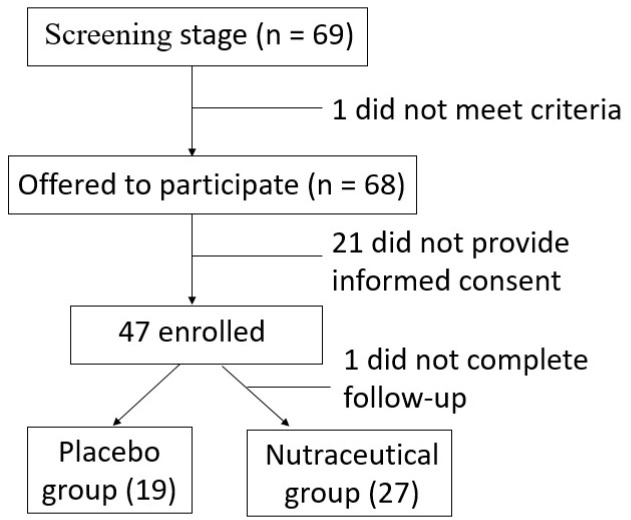
Flow diagram showing patient inclusion.

**Figure 3 biomedicines-13-01185-f003:**
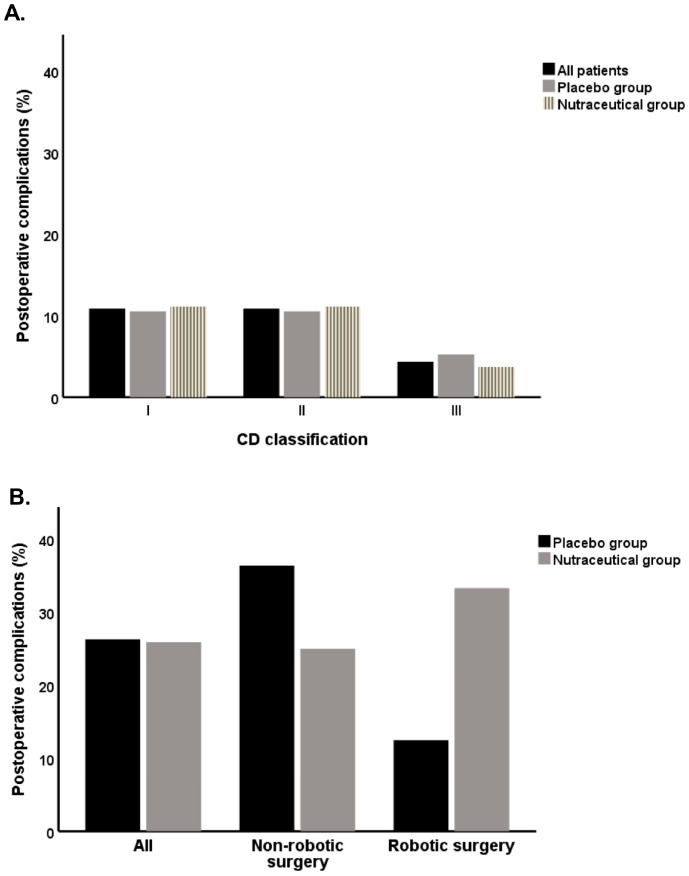
Rate of postoperative complications in the placebo group and nutraceutical group of patients: (**A**) CD classification, All patients: CD I n = 5, CD II n = 5, CD III n = 2; Placebo: CD I n = 2, CD II n = 2, CD III n = 1; Nutraceutical: CD I n = 3, CD II n = 3, CD III n = 1; (**B**) postoperative complications in non-robotic and robotic surgery patients, All: Placebo n = 5, Nutraceutical n = 7; Non-robotic surgery: Placebo n = 4, Nutraceutical n = 6; Robotic surgery: Placebo n = 1, Nutraceutical n = 1.

**Figure 4 biomedicines-13-01185-f004:**
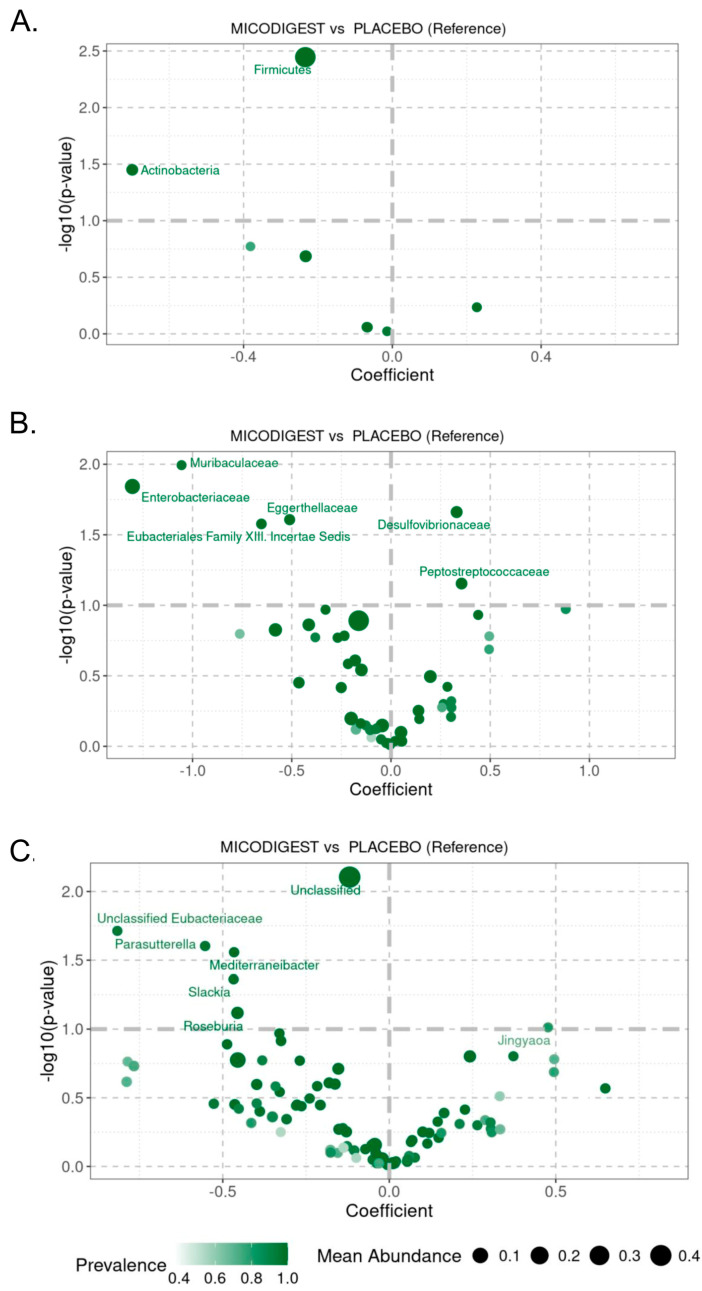
Volcano plot showing *p*-values of linear regression for LinDA at (**A**) phylum, (**B**) family, and (**C**) genus level between placebo and nutraceutical (MICODIGEST) groups of patients.

**Table 1 biomedicines-13-01185-t001:** Baseline characteristics of the patients.

	Placebo Group (n = 19)	Nutraceutical Group (n = 27)	*p* ^1^
Age, (years)	68.0 (63.0–73.0)	67 (62.3–72.8)	0.4
Gender, male/female	11/8	12/15	0.4
Tumor location, n (%)			0.4
Distal colon	10 (52.6)	15 (55.6)	
Proximal colon	5 (26.3)	10 (37.0)	
Rectum	4 (21.1)	2 (7.4)	
CRC diagnosis, n (%)			0.6
Positive FIT	10 (52.6)	13 (48.1)	
Surveillance	3 (15.8)	3 (11.2)	
Symptoms	6 (31.6)	9 (33.4)	
Others	0	2 (7.4)	
Duration of symptoms (weeks)	0 (0–15.0)	0 (0–14.8)	0.5
BMI (Kg/m^2^)	25.3 (23.6–30.1)	26.0 (23.5–30.2)	0.7
Fat mass (%)	30.8 (24.7–37.7)	31.8 (24.5–40.3)	0.4
Muscle mass (%)	65.7 (59.1–71.1)	64.6 (56.7–70.1)	0.5
Hemoglobin (g/dL)	13.7 (12.5–14.5)	13.7 (12.6–14.3)	0.3
White blood cells (10^3^/µL)	6200 (5200–6970)	6020 (5180–6740)	0.7
Lymphocytes (10^3^/µL)	1590 (1233–1830)	1441 (1090–1755)	0.4
Neutrophils (10^3^/µL)	3680 (3128–4868)	3690 (2875–4555)	0.6
Neutrophil/Lymphocyte ratio	2.63 (2.24–3.08)	2.49 (2.06–3.47)	0.9
Creatinine (mg(dL)	0.9 (0.7–0.9)	0.8 (0.7–0.9)	0.4
Albumin (g/dL)	4.5 (4.3–4.6)	4.4 (4.3–4.5)	0.3
Prothrombin time (seg)	10.8 (10.5–11.4)	10.7 (9.9–11.1)	0.1
IL-6 (pg/mL)	3.6 (2.3–4.8)	3.0 (2.5–4.7)	0.6
IL-10 < 1.6 pg/mL, n (%)	18 (94.7)	27 (100)	0.2
TNF-α (pg/mL)	9.0 (7.6–11.3)	6.0 (7.0–11.1)	0.1

^1^ Differences between qualitative variables were analyzed with the chi-square test. Differences between quantitative variables were analyzed with Student’s *t*-test or the Wilcoxon test if variables did not meet normality. Differences with *p* < 0.05 are considered statistically significant. Continuous variables were presented as medians and interquartile range (IQR). Categorical variables were expressed as frequencies and percentages. FIT: fecal immunochemical test, BMI: body mass index, IL-6: interleukin-6, IL-10: interleukin-10, TNF-α: tumor necrosis factor alpha.

**Table 2 biomedicines-13-01185-t002:** Duration of treatment, adverse events, surgery, and postoperative complications.

	Placebo Group (n = 19)	Nutraceutical Group (n = 27)	*p* ^1^
Duration of treatment (weeks)	3 (4–3)	3 (4–2)	0.9
Adherence to treatment (>80%)	18 (94.7)	24 (88.9)	0.5
Adverse events	8 (42.1)	7 (25.9)	0.2
Adverse events (≥III)	0 (0)	1 (14.3)	0.2
Type of surgery:			
Right hemicolectomy	5 (26.3)	9 (33.3)	0.6
Sigmoidectomy	5 (26.3)	9 (33.2)	0.6
Left hemicolectomy	0 (0)	2 (7.4)	0.2
Low anterior resection	4 (2.1)	1 (3.7)	0.1
Rectosigmoid resection	3 (15.8)	1 (3.7)	0.2
Segmental colectomy	2 (10.5)	1 (3.7)	0.4
Other	0 (0)	4 (14.8)	0.1
Surgical approach:			
Laparoscopy	9 (47.4)	20 (74.1)	0.07
Conversion to open surgery	2 (10.5)	2 (7.4)	0.7
Open surgery	0 (0)	2 (7.4)	0.0
Robotic assisted surgery	8 (42.1)	3 (11.1)	0.02
Postoperative complications			
I	2 (10.5)	3 (11.)	0.9
II	2 (10.5)	3 (11.1)	0.9
III	1 (5.3)	1 (3.7)	0.8

^1^ Differences between qualitative variables were analyzed with the chi-square test. Differences between quantitative variables were analyzed with Student’s *t*-test or the Wilcoxon test if variables did not meet normality. Differences with *p* < 0.05 are considered statistically significant. Continuous variables were presented as medians and interquartile range (IQR). Categorical variables were expressed as frequencies and percentages. Adverse events and their severity were documented according to the Common Terminology Criteria for Adverse Events (CTCAE), version 5.0. Postoperative complications were evaluated using the Clavien-Dindo classification.

**Table 3 biomedicines-13-01185-t003:** Comparison of nutritional status, quality of life, and blood parameters between the placebo group and the nutraceutical group at the end of follow-up.

	Placebo Group (n = 19)	Nutraceutical Group (n = 27)	*p* ^1^
BMI (Kg/m^2^)	26.2 (23.6–29.7)	26.0 (23.5–30.1)	0.8
Fat mass (%)	30.1 (23.6–37.4)	30.9 (23.6–38.9)	0.6
Muscle mass (%)	66.04 (59.4–72.7)	65.6 (58.1–72.7)	0.3
Quality of life:			
Physical functioning	100 (90–100)	100 (77.5–100)	0.4
Physical role	100 (100–100)	100(100–100)	1.0
Emotional role	100 (66.7–100)	100 (66.7–100)	0.8
Energy/vitality	70 (55–90)	70 (56.3–85)	0.8
Mental health	80 (52–96)	80 (61–95)	1.0
Social functioning	100 (75–100)	100 (87.5–100)	0.8
Bodily pain	100 (80–100)	100 (70–100)	0.5
General health perceptions	60 (45–80)	65 (46.3–80)	0.4
Blood parameters:			
Hemoglobin (g/dL)	13.3 (12.8–13.95)	13.3 (12.7–13.9)	0.5
White blood cells (10^3^/µL)	6160 (5290–7185)	5990 (4650–7130)	0.7
Lymphocytes (10^3^/µL)	1435 (1093–1780)	1730 (1105–2090)	0.3
Neutrophils (10^3^/µL)	4200 (2978–4840)	3590 (2285–4415)	0.1
Neutrophil/Lymphocyte ratio	2.62 (2.04–3.58)	2.06 (1.29–2.86)	0.07
Creatinine (mg(dL)	0.9 (0.8–1.0)	0.8 (0.8–1.0)	0.6
Albumin (g/dL)	4.3 (4.2–4.5)	4.2 (4.1–4.4)	0.1
Prothrombin time (seg)	11.0 (10.3–11.7)	9.2 (7.5–10.5)	0.1
IL-6 (pg/mL)	3.6 (2.4–5.9)	3.5 (2.7–5.2)	0.5
IL-10 < 1.6 pg/mL, n (%)	17 (89.5)	25 (92.6)	0.7
TNF-α (pg/mL)	9.0 (7.6–12.1)	6.0 (7.5–10.5)	0.9

^1^ Differences between qualitative variables were analyzed with the chi-square test. Differences between quantitative variables were analyzed with Student’s *t*-test or the Wilcoxon test if variables did not meet normality. Differences with *p* < 0.05 are considered statistically significant. Continuous variables were presented as medians and interquartile range (IQR). Categorical variables were expressed as frequencies and percentages. FIT: fecal immunochemical test, BMI: body mass index, IL-6: interleukin-6, IL-10: interleukin-10, TNF-α: tumor necrosis factor alpha.

## Data Availability

The original contributions presented in this study are included in the article/Appendix A. Further inquiries can be directed to the corresponding author.

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
