# Peer review of "The Effect of Fungal Nutraceutical Supplementation on Postoperative Complications, Inflammatory Factors and Fecal Microbiota in Patients Undergoing Colorectal Cancer Surgery with Curative Intent: A Randomized, Placebo-Controlled, Double-Blind Clinical Trial"

_biomedicines, 2025, doi:10.3390/biomedicines13051185_

Round 1

Reviewer 1 Report

Comments and Suggestions for Authors

Summary: In this manuscript, Dr. Reguiero and colleagues discuss the results of a clinical trial testing the ability of MicoDigest 2.0 to reduce post-operative complications in patients undergoing resection for colorectal cancer. Unfortunately, only 46 of the 144 needed patients consented to this study, leaving it grossly underpowered. Given the lack of power, none of the primary outcomes were different between the intervention and placebo arms. The only differences found between treatment groups are in a subgroup analysis of bloodwork, evaluating only the non-robotic surgical cases, and some of the specific microbial changes in the pre- and post-intervention samples. These differences may be interesting, but there a couple things that need to be addressed before this manuscript can be reconsidered:

  1. Throughout the manuscript, the results are presented a bit too optimistically (see lines 254-260 on page 7). A more straightforward representation of these results would be ‘no significant associations were found after subgroup analysis’.
  2. The first statistically significant finding is in the subgroup analysis, wherein the authors only evaluated patients undergoing non-robotic surgery. Although this is the majority of the patients, the differences in the lymphocytes and NLR seems to be driven by lower lymphocytes in the placebo group (compare tables 2 and Supplementary table 2). This needs to be discussed. Is this likely to be reproducible?
  3. The second statistically significant finding is in the bacterial composition post-treatment. While interesting, there is a surprising lack of discussion as to the potential meaning of these differences. From the literature, are such changes consistent with better outcomes, or in an unexpected direction?
  4. Given the sample size and number of analyses, is it reasonable to use an unadjusted p-value of 0.05 (pg 5, line 179)?
  5. The discussion is under-developed. There is a fair bit of text spent trying to ‘sell’ the reader on the viability of fungal supplements for CRC outcomes, but no discussion on the relevance of the differences found in bacterial species or other important points. Why so much trouble recruiting patients? If more research is warranted (not clear to me at this point), will the same recruitment problems persist?

Author Response

Response to Reviewer 1

Comment 1: Throughout the manuscript, the results are presented a bit too optimistically (see lines 254-260 on page 7). A more straightforward representation of these results would be ‘no significant associations were found after subgroup analysis’.

Response: We agree. We have changed the description of the findings mainly those related to the surgical complications (see 3.2).

Comment 2: The first statistically significant finding is in the subgroup analysis, wherein the authors only evaluated patients undergoing non-robotic surgery. Although this is the majority of the patients, the differences in the lymphocytes and NLR seems to be driven by lower lymphocytes in the placebo group (compare tables 2 and Supplementary table 2). This needs to be discussed. Is this likely to be reproducible?

Response: Thank you for pointing this out. This comment was also made by the second reviewer and we have added some specific comments in the discussion based on a recently publishesdsystematic review that evaluates its relation with the outcomes after surgical resection of CRC.

Comment 3: The second statistically significant finding is in the bacterial composition post-treatment. While interesting, there is a surprising lack of discussion as to the potential meaning of these differences. From the literature, are such changes consistent with better outcomes, or in an unexpected direction?

Response: We have now expanded the Discussion to explore the potential implications of the observed microbial shifts. We have added a reference and a new paragraph in the discussion

Comment 4: Given the sample size and number of analyses, is it reasonable to use an unadjusted p-value of 0.05 (pg 5, line 179)?

Response: We acknowledge the limitation. We now clearly state in the Methods and Discussion that no correction for multiple comparisons was applied, and results should be interpreted as exploratory. This caveat is included on section 2.7.

Comment 5: The discussion is under-developed. There is a fair bit of text spent trying to ‘sell’ the reader on the viability of fungal supplements for CRC outcomes, but no discussion on the relevance of the differences found in bacterial species or other important points. Why so much trouble recruiting patients? If more research is warranted (not clear to me at this point), will the same recruitment problems persist?

We thank the reviewer for this comment. We are aware of the methodological limitations of the results presented, due to the difficulties in reaching the estimated number of patients. Conducting a clinical trial is not an easy task, and in this case, the challenges were multiple. The main one was that patients did not perceive a potential benefit from participating in the trial. We believe these limitations are clearly stated in the discussion section of the manuscript. On the other hand, we think we are opening a door that deserves further exploration in future projects, as we have identified a pathophysiological basis that may justify potential benefits for patients. In this regard, and in line with the reviewers’ comments, we have expanded two points in the discussion: the neutrophil-to-lymphocyte ratio and the potential impact of phyla on the prognosis of patients with colorectal cancer

Reviewer 2 Report

Comments and Suggestions for Authors

Congratulations to the research team.  Setting up a clinical trial is quite difficult.  

In this study, it was found that many of the patients who initially gave their consent later refused to be part of the study. This has resulted in a small number of participants.  I encourage you to continue to increase the sample size and check that the results are the same.

From my point of view, it can be improved if there would be some changes in the following points:

  1. The Section 3.2. Association of Nutraceutical Supplementation with Postoperative Complications, is quite difficult to read and understand. Is it possible to summarise it in a table? .
  2. Line 237: What are the adverse effects?
  3. In section 3.4, lines 297-301, it is indicated that there is a difference between the samples collected at the beginning and those collected at the end. What is the reason for this difference, as it is not explained in the text?.
  4. Lines 370-372. The discussion suggests that there is no references to support the outcome ratio of neutrophils to lymphocytes in relation to CRC. This ratio has been observed to be associated with different stages of CRC, which is not reported in this study.

Author Response

Response to Reviewer 2

Comment 1: The Section 3.2. Association of Nutraceutical Supplementation with Postoperative Complications, is quite difficult to read and understand. Is it possible to summarize it in a table?

Response: Thanks for the suggestion. We have included a table (2) to summarize all the information regarding adherence, adverse events related to treatment, surgery and, finally, postoperative surgical complications. Furthermore, we have divided the paragraph into three parts to make it easier to read.

Comment 2: Line 237: What are the adverse effects?

Response: We did not include the information regarding to each type of adverse events. We classified them during treatment according to the Common Terminology Criteria for Adverse Events (CTCAE), version 5.0. In any case, we found no differences between each arm and a low rate of serious adverse events.

Comment 3: In section 3.4, lines 297-301, it is indicated that there is a difference between the samples collected at the beginning and those collected at the end. What is the reason for this difference, as it is not explained in the text?

Response: We agree that it is not correctly stated. At the time of inclusion, 45 patients provided a sample, and of these, 36 patients provided a second sample at the end of the treatment. We changed it in the results section.

Comment 4: Lines 370-372. The discussion suggests that there is no references to support the outcome ratio of neutrophils to lymphocytes in relation to CRC. This ratio has been observed to be associated with different stages of CRC, which is not reported in this study.

Response: We have now added references linking NLR to CRC. We have identified a systematic review recently published that associated NLR with complications and overall survival (Shevchenko, I.; Serban, D.; Simion, L.; Motofei, I.; Cristea, B.M.; Dumitrescu, D.; Tudor, C.; Dascalu, A.M.; Serboiu, C.; Tribus, L.C.; et al. Clinical Significance of Blood Cell-Derived Inflammation Markers in Assessing Potential Early and Late Postoperative Complications in Patients with Colorectal Cancer: A Systematic Review. J. Clin. Med. 2025, 14, 2529. https://doi.org/10.3390/jcm14072529) We have changed the discussion accordingly.

Round 2

Reviewer 1 Report

Comments and Suggestions for Authors

Thank you for your attention to the prior review comments. In addition to citing the study on pre-operative neutrophil-to-lymphocyte ratio in the discussion, it would be helpful to point out why this reference is relevant to your results. Specifically the difference in pre-operative levels in your subgroup analyses.

Author Response

Comment 1: Thank you for your attention to the prior review comments. In addition to citing the study on pre-operative neutrophil-to-lymphocyte ratio in the discussion, it would be helpful to point out why this reference is relevant to your results. Specifically the difference in pre-operative levels in your subgroup analyses.

Response: We have added the following comment: “In our study, we identified a non-significant reduction in the ratio across the entire patient cohort, which reached statistical significance in patients who underwent non-robotic surgery, and thus experienced greater surgical trauma. Our results are consistent with previously published data. However, we were unable to confirm an association with clinically relevant outcomes, and therefore further research is needed to determine whether this ratio can serve as a predictor of postoperative evolution following colorectal surgery”.
